# One Leaf Knows Autumn: A Piece of Data-Model Facilitates Efficient Cancer Prognosis with Histological and Genomic Modalities

## Abstract

The rapidly emerging field of computational pathology enables integrated image-omic solutions for cancer prognosis by jointly modeling both histological and genomic data. However, current multi-modal techniques suffer from three major bottlenecks: (1) Memory Overheads, since a raw histology image typically has a super high resolution, e.g., $203,183 \times 91,757$ in cancer `HNSC`. Simple patch partitioning trades training time for spaces. (2) Massive Computing Costs, due to immense parameter counts in recent state-of-the-art models, which demand substantial computational resources. Meanwhile, their intrinsic representation redundancy in vanilla-trained networks leads to an ineffective usage of the capacity. (3) Gradient Conflicts, because there are significant heterogeneities between image and genomic data modalities, resulting in the disagreement of optimization directions. In this work, we propose an effective multi-modal pipeline for cancer prognosis, i.e., `CancerMoE`, to address the aforementioned challenges. Specifically, from data to model, it first designs a dynamic patch selection algorithm to flexibly score and locate informative patches online, trimming down the memory cost; then introduces a Sparse Mixture-of-Experts (SMoE) framework to disentangle weight spaces and allocate the most relevant model pieces to an input sample, promoting training efficiency and synergistic optimization among multiple modalities; finally, consolidates and scarifies redundant attention heads, leading to improved efficiency and interpretability. Our extensive experiments demonstrate that `CancerMoE` achieves competitive performance on **twelve** cancer datasets compared to previous methods. Meanwhile, our proposed network architecture requires only **1%** of the image patches, **20%** of the model parameters, and **30%** of the merged attentions compared with the vanilla transformer network. Key codes are provided in detail in the supplement.

## 1 Introduction

In cancer research, a comprehensive examination of various facets is often needed to unravel the intricate nature of this complex disease Marusyk & Polyak (2010); Marusyk et al. (2012). Prognosis Sala et al. (2017); Thakor & Gambhir (2013) serves as one of the promising approaches to develop an understanding of cancer and predict the survival chance of patients, equipping with cutting-edge technologies like molecular profiling Yanaihara et al. (2006), imaging modalities Shahbazi-Gahrouei et al. (2019), and genetic analysis Kamps et al. (2017); Claus et al. (1991). Moreover, the joint investigation between tumor microenvironments ( *e.g.,* histological images) and its interplay with immune responses ( *e.g.,* genomic profiles) sheds light on the intrinsic dynamics that influence tumor development and metastasis Heindl et al. (2015); Kather et al. (2018); Tarantino et al. (2021), paving the way for effective survival prediction and further treatment.

Specifically, the advent of high-throughput sequencing technologies has brought about significant advances in survival analysis, leading to a shift from the sole examination of clinical indicators to the integration of genomic profiles with pathological images. Recent investigations Shimizu et al. (2022); Gobin et al. (2019); Kalra et al. (2020); Mayekar & Bivona (2017); Zhang et al. (2022); Lu et al. (2022) have highlighted the benefits of exploring multi-modal analysis. Unfortunately, current learning-based integration solutions are

still in the initial stage of fusing multi-modal knowledge in a straightforward way. For instance, Braman et al. (2021); Cheerla & Gevaert (2019); Chen et al. (2020) directly combine pathological features and genomic profiles for survival prediction, which overlook inherent cross-modal interactions; Li et al. (2021); Wang et al. (2021); Chen et al. (2021b) utilize genomic embeddings to guide the attention aggregation of pathological image features, disregarding information that may not be associated with gene expressions. Thus, there is an immediate demand for an effective integration mechanism adept at deciphering the domain-specific heterogeneity within histological and genomic data modalities. Concurrently, while recent advancements in learning algorithms have shown promising performance that surpasses human experts, their demanding computational cost poses significant challenges to scalability and practical application.

In light of this, our paper targets effective integration, aiming to address computing bottlenecks in three intertwined aspects. ① *Memory Overheads.* Histology images usually have super high resolutions, *e.g.*, $191,352 \times 91,562$ in cancer `LUAD` and $139,008 \times 256,256$ in cancer `BRCA`, which require substantial CPU/GPU memories to load and process the data. A conventional way is segmenting the high-resolution images and creating millions of smaller patches Kong et al. (2023); Dosovitskiy et al. (2020). However, it actually trades longer training time for memory reductions. ② *Training Efficiency.* Millions of patches and huge parameter counts in recent State-of-the-Art (SoTA) transformer-based models Chen et al. (2022a) severely question the resource intensity during training. ③ *Inference Efficiency.* Another efficiency concern and drawback lies in the insufficient utilization of model capacity. As presented in recent studies Yuan et al. (2021); Gao et al. (2021); He et al. (2023), only a small portion of network weights, like 5% Zhang et al. (2021); Allen-Zhu & Li (2019) of total parameter counts, are engaged during the inference of each sample. A few pioneering efforts have explored dynamic sparsity as initial remedies, to cut redundancy and boost training and inference efficiency.

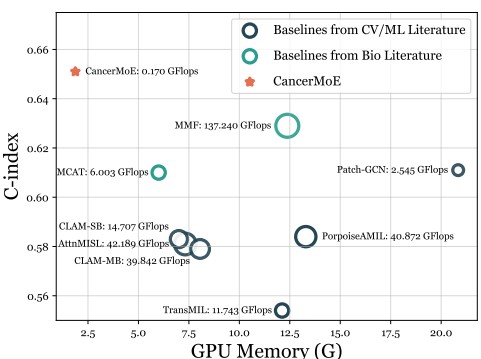

Figure 1: The performance of histology-genomic cancer prognosis on the `BRCA` dataset. Averaged results and the required memory cost are reported. Markers ⋆ and ⭕ denote "ours" and "baseline" approaches, respectively. A larger marker indicates more floating point operations (FLOPs) are used for inference. The top left corner implies the most ideal solution.

To overcome the aforementioned challenges in terms of effective integration and efficient computing (①+②+③), we propose a novel framework, namely, `CancerMoE`, for ultra-efficient data integration for cancer prognosis. The innovative designs span both data and model perspectives. Along with the information feedforward, it first employs a dedicated Dynamic Patch Selector (DPS) that meticulously examines and selects crucial image patches abundant in histological information, while discarding redundant ones. It significantly reduces the heavy costs associated with memory and training time. Then, a tailored SMoE architecture is invented to learn modality-*specific* and *-agnostic* modules to synergize multi-modality optimization. In detail, modularization and modality-aware routing policies are leveraged to disentangle the model parameter space and allocate input tokens to different model pieces, aiming for computational efficiency and mitigated conflicts of multi-modal gradients, respectively. Lastly, we investigate and diminish the attention redundancy by proposing the Attention Consolidation and Sparsification (ACS) mechanism. It appropriately clusters multiple attention heads and reduces superfluous attention connections, which brings improved training and inference efficiency and interpretability. Our innovation efforts can be summarized into the following four thrusts:

⋆ Given 1% patches of histological images, the same genomic profiles, and 20% model parameters, we demonstrate promising performance and efficiency for predicting the survival of cancer patients. We introduce `CancerMoE`, an effective multi-modal learner in cancer prognosis, that seamlessly integrates histology images and genomics profiles.

⋆ We design a dynamic patch selector mechanism to score and select the most crucial image patches (*e.g.*, 1% of total patches) online, which avoids loading the full-resolution image and discards its massive memory overheads.

⋆ We propose a consolidation and sparsification algorithm for self-attention modules to reduce intrinsic redundancy and promote efficiency. It first merges insignificant attention heads into a few knowledgeable ones, then eliminates less informative elements in their attention maps.

⋆ Extensive empirical studies are conducted to validate the effectiveness of `CancerMoE` on **twelve** representative cancer datasets. Specifically, our approaches surpass the **ten** existing state-of-the-art methods by a clear performance margin of $4.1\% \sim 18.2\%$ accuracies with $8.9\% \sim 31.2\%$ memory and $0.1\% \sim 2.1\%$ FLOPs as shown in Fig. 1.

## 2 Related Work

**Multi-Modality Learning (MML).**  Integrating multiple modalities of data (*e.g.*, vision, text, and audio) has a long history in the Machine Learning (ML) community Lahat et al. (2015); Bayoudh et al. (2021); Ngiam et al. (2011); Baltrušaitis et al. (2018). The recent trend is leveraging transformer-based models as a universal backbone to enable effective multi-modal learning Ramesh et al. (2022); Saharia et al. (2022); Xia et al. (2023); Dai et al. (2022). MML also plays a crucial role in medical applications. For example, Suresh et al. (2017) merge chest X-rays, textual clinical notes, and longitudinal measurements for intensive care monitoring. Zhou & Chen (2023) design a cross-modal translation and alignment framework to capture intrinsic cross-modal correlation and discard irrelevant pathological information to gene expressions. Nowadays, along with the rapid and explosive developments of computing and AI4Medicine, there has been a surge in research efforts aimed at creating systems proficient in multimodal medical scenarios Subbiah Parvathy et al. (2020); Huang et al. (2023); Zhu et al. (2022b); Muhammad et al. (2021); Li et al. (2022). Among this huge family, Zhu et al. (2022b) propose an adaptive co-occurrence pipeline to improve the modality fusion quality through a filter-based image decomposition algorithm. Subbiah Parvathy et al. (2020) utilizes an enhanced monarch butterfly optimization to decide a better threshold of fusion rules in shearlet transform, where low and high-frequency sub-bands are fused on the basis of corresponding feature maps. Li et al. (2022) presents a hierarchical multimodal integration approach by employing a factorized bilinear model to fuse genomic and image features in a step-by-step manner.

**Histology-Genomic Cancer Prognosis.**  Cancer prognosis involving histological images and genomic data has gained increasing popularity Chen et al. (2020); Li et al. (2022), which seamlessly combines tissue structure study with genetic data analysis. For instance, Galateau-Salle et al. (2016) explores an integrative genomics framework for constructing a prognostic model to clear renal cell carcinoma. Hao et al. (2019) introduces a biology-informed ML pipeline to identify genetic and histopathological patterns, aiming at advanced survival predictions. Mobadersany et al. (2018) shows superior prediction accuracy of patients' overall survival. To be more specific, they create a multi-modal clinical paradigm to learn and integrate knowledge from both histology images and genomic biomarkers. Recently, numerous efforts Natrajan et al. (2016); Kather et al. (2019); Coudray et al. (2018); Subramanian et al. (2018); Mobadersany et al. (2018); Echle et al. (2021); Hou et al. (2022) have been conducted to revolutionize how we diagnose and treat cancer by jointly modeling both tissue structure (*e.g.*, pathological images) and genetic data.

**Sparse Mixture-of-Experts (SMoE).**  The conventional dense mixture-of-experts utilizes all experts for each input sample, and therefore it is computationally demanding and expensive. Latest investigations Lepikhin et al. (2020); Shazeer et al. (2017a); Fedus et al. (2022) propose an efficient alternative, *i.e.*, SMoE, by sparsely activating a small portion of experts. Such sparse gating fashion brings substantial efficient gains in both the training and inference stages, facilitating the scaling of deep neural networks to massive sizes, *e.g.*, even to trillions of parameters Fedus et al. (2022). SMoEs have exhibited remarkable efficacy in the realms of computer vision Lou et al. (2021); Eigen et al. (2013); Riquelme et al. (2021); Ahmed et al. (2016); Gross et al. (2017); Wang et al. (2020); Abbas & Andreopoulos (2020); Pavlitskaya et al. (2020) and NLP Kim et al. (2021b); Shazeer et al. (2017a); Lepikhin et al. (2020); Zhou et al. (2022); Zhang et al. (2021); Zuo et al. (2021); Jiang et al. (2021). Several pioneering investigations take the advantage of this conditional computing nature to allocate task-relevant Ma et al. (2018); Aoki et al. (2021); Hazimeh et al. (2021); Kim et al. (2021a); Fan et al. (2022); Ye & Xu (2023); Chen et al. (2023) model pieces or modality-relevant Mustafa et al. (2022); Zhu et al. (2022a); Kudugunta et al. (2021) ones in a dynamic way for multi-task or multi-modal learning, respectively. In cancer research, a few pilot studies have been performed Raman et al. (2010); Myoung

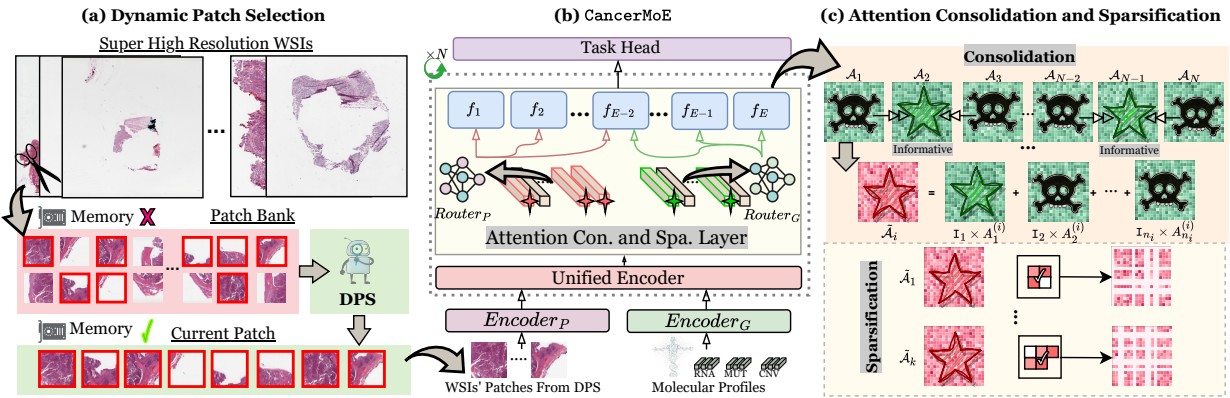

Figure 2: The entire architecture design of `CancerMoE`. The key components of this network include: (1) Dynamic Patch Selection (DPS) flexibly scores all patches in an online fashion, identifying elite patches; (2) Histopathological images and genomic data are individually transformed into embeddings and merged across modalities within a unified encoder; (3) Leveraging the Attention Consolidation and Sparsification (ACS) mechanism, the `CancerMoE` automatically filters out elements with low informational value from attention maps, selectively guiding high-quality tokens to respective experts for efficient cancer prognosis prediction.

(2013); Übeyli (2005); Kreutz et al. (2001); Afshar et al. (2021), but most investigations are limited to single modality learning and single cancer. The significant heterogeneity among modalities, memory bottlenecks gave super-high-resolution images, and diverse learning objectives led to challenging optimization issues for SMoE models and their study in cancer prognosis problems. In this regard, this paper attempts to provide an effective answer.

## 3 Methodology

**Revisting Sparse Mixture-of-Experts (SMoE).** The SMoE pipeline typically contains a router $\mathcal{R}$ and a group of experts $\{f_1, f_2, \ldots, f_E\}$, where E is the number of experts. The output representation is then calculated by $\boldsymbol{y} = \sum_{i=1}^{E} \mathcal{R}(\boldsymbol{x})_i \cdot f_i(\boldsymbol{x})$, where $f_i(\boldsymbol{x})$ denotes the intermediate feature produced by expert $f_i$ and a weighted summation is performed based on their coefficients $\mathcal{R}(\boldsymbol{x})_i$. Specifically, the router function is described as $\mathcal{R}(\boldsymbol{x}) = \texttt{TopK}(\texttt{softmax}(g(\boldsymbol{x})), k)$, where $\mathcal{R}$ activates the top-$k$ expert networks with the largest scores $g(\boldsymbol{x})$ given an input embedding $\boldsymbol{x}$. Also, $g$ is a learnable neural network, as a Multi-Layer Perception (MLP). Meanwhile, the `TopK` function is shown as:

$$\texttt{TopK}(\boldsymbol{v}, k) = \begin{cases} \boldsymbol{v} & \textit{if } \boldsymbol{v} \textit{ is in the top } k \\ 0 & \textit{otherwise} \end{cases}, \tag{1}$$

which preserves the largest $k$ values in $\boldsymbol{v}$ and sets the rest of the elements to zero.

**Revisting Self-Attention.** In the classic design of a self-attention mechanism, input tokens $\{p_i\}_{i=1}^{L}, p_i \in \mathbb{R}^{d \times 1}$ are fed into three linear layers to produce the query $\mathcal{Q}$, key $\mathcal{K}$, and value $\mathcal{V}$ matrices, respectively. Each output matrix, $\mathcal{Q}, \mathcal{K}, \mathcal{V} \in \mathbb{R}^{L \times d}$ shares a hidden dimension $d$, with L being the number of total tokens. The attention module `Attn` is then formulated as $\texttt{Attn}(\mathcal{Q}, \mathcal{K}, \mathcal{V}) = \texttt{softmax}(\frac{\mathcal{Q}\mathcal{K}^\top}{\sqrt{d}})\mathcal{V}$. To be specific, let $\mathcal{Q} = [q_1, q_2, \cdots, q_L]$ and $a_i = \texttt{softmax}(\frac{q_i \mathcal{K}^\top}{\sqrt{d}}) \in \mathbb{R}^L$ is the attention scores for the $i$th token $p_i$. As for the multi-head self-attention, $\mathcal{H}$ self-attention modules are applied to $\{p_i\}_{i=1}^{L}$ separately, and a weighted averaging is then performed on top of their outputs to generate the final representation. The corresponding attention score is modified as $\tilde{a}_i = \frac{1}{\mathcal{H}} \sum_{h=1}^{\mathcal{H}} a_i^h$.

### 3.1 `CancerMoE` - **An Ultra-Efficient Multi-Modal Integration Framework in Cancer Prognosis**

**Overview of `CancerMoE`.** `CancerMoE` is a multi-modal integration algorithm that learns and infers from histology and genomics information for cancer prognosis. Together with a tailored SMoE architecture, two

efficient designs are proposed from data (*i.e.,* dynamic patch selection in Section 3.3) and model (*i.e.,* attention consolidation and sparsification in Section 3.4) perspectives, aiming for fast cancer prognosis. The overall procedures of `CancerMoE` are illustrated in Fig. 2. It first selects the most influential histological image patches in a data-driven manner. Then, it turns all raw modalities into embeddings and feeds them into a unified transformer encoder to fuse the knowledge across modalities. Finally, all token embeddings are passed through our customized SMoE equipped with consolidated and sparsified attention modules. After this step, these tokens are fed to corresponding experts via modality-specific routing for cancer prognosis prediction.

**Customized SMoE Architecture.** In this work, we focus on transformed-based networks since they have demonstrated numerous successes in unifying heterogeneous modalities Zhu et al. (2022a). Our tailored designs span two aspects: ① *Modality-Specific Embedding and Routing Policies.* `CancerMoE` creates modality-specific embedding by concatenating the one-hot modality index vector and the token embedding as $\boldsymbol{x}_m = \texttt{Concat}(\boldsymbol{x}, \texttt{OneHot}(m))$, where $\boldsymbol{x}$ and $\texttt{OneHot}(m)$ denote the intermediate token embedding and its one-hot index vector of the modality $m$, respectively. On top of $\boldsymbol{x}_m$, modality-aware routing is enabled according to $\mathcal{R}(\boldsymbol{x}_m)$. The design philosophy is to encourage a synergized multi-modal optimization by learning appropriate modality-*specific* and -*agnostic* expert assignments, which provides possibilities to uncover hidden cross-modality interactions and transcends the capabilities of any single modality, as demonstrated in Sec. 4.4.

② *Modularization.* For efficiency purposes, we turn a large, densely connected model into the mixture-of-experts architecture. Specifically, a uniform partition is adopted to divide the original MLP into multiple smaller MLPs. Without loss of generality, let $d$ be the dimensionality of the original MLP. After our modularization, a series of MLP experts $\{f_1, f_2, \cdots, f_E\}$ is obtained with the same hidden dimension $\frac{d}{E}$. Note that, at both training and inference phases, only a small subset of experts are activated for the prediction of one sample, facilitating efficient cancer prognosis (Table 2). Meanwhile, such model division allows a disentanglement in the model parameter space, offering opportunities to mitigate conflicted gradient directions from diverse modalities(Figure 5 **(b)**).

## 3.2 Genomic Profile Encoder

To integrate genomic information, we use "PatchEmbedding" to encode the genomic profiles. Specifically, we start by treating the genomic profiles as a single vector, which we divide into $g$ sub-vectors. Each sub-vector is then projected into the embedding space through a linear layer. After this, we concatenate the sequence of genomic profile tokens with the image tokens to create a single input sequence. This combined sequence is then processed by the transformer backbone, where the self-attention modules merge the two types of data.

## 3.3 Dynamic Patch Selection for Cancer Images with Super High Resolutions

The fine-gained histological image information is necessary for prognosis Shaban et al. (2019); Kim et al. (2020). Nevertheless, there are two challenges that hinder the effective and efficient utilization of this special modality for prognosis: (1) the super-high-resolution Whole Slide Images (WSIs) result in unbearable computation costs; (2) the interfering noise level increases with the image resolution.

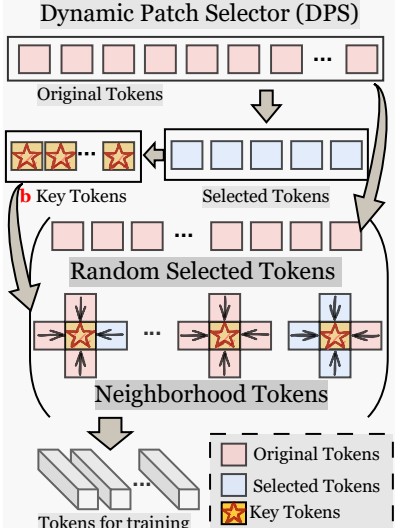

Figure 3: The procedure of Dynamic Patch Selector (DPS).

To tackle these issues, we present the Dynamic Patch Selector (DPS) framework. The DPS begins by segmenting all whole slide images (WSIs) into patches and storing them in the patch bank for each example. Partial patches then undergo a dynamic scoring process, through which a small subset of the most informative patches, deemed worthy of learning, is selected. Simultaneously, a random subset of the remaining patches within the patch bank is also chosen to prevent overfitting and explore other informative patches. Subsequently, the chosen patches are collaboratively used for the online training of our proposed `CancerMoE` framework, resulting in significant training cost reductions and effective noise token filtration.

**Proposed Remedies of Addressing the Redundancy Issues to Recover Efficiency.** Recently, Rao et al. (2021); Kong et al. (2021) have observed the information contained in tokens has diverse ranges, which indicates that there are redundant and noisy tokens among the WSIs. We can remove less informative tokens to save computational costs and filter noisy tokens to achieve superior performance.

Our policy for managing computational costs involves reviewing a fixed number of WSI tokens during each training iteration. This subset comprises two distinct parts: neighborhood tokens and randomly selected tokens. As depicted in Figure 3, tokens processed by the network will be assigned a token score, hereafter referred to as "Selected Tokens". The DPS first retrieves $b$ tokens with the highest token scores, designated as "Key Tokens", which treats them as informative tokens. Intuitively, informative tokens are not isolated, regions surrounding "Key Tokens" are likely to contain pertinent information. For instance, the extent of cancerous tissue often surpasses the size of a single patch (each patch will be processed as a token). Hence, we select tokens centered on "Key Tokens" as neighborhood tokens. However, in the initial stages of training, the token scores from the attention mechanism could be inaccurate, potentially leading to network overfitting on less informative tokens. To mitigate this, the DPS concurrently retrieves tokens from the "Original Tokens" at random- termed "Random Selected Tokens". This parallel selection process explores new informative tokens and prevents overfitting to initially favored yet sub-optimal tokens.

**Token score from self attention.**, for token score assignment, we introduce the unique learnable token, denoted as $x_{\mathrm{cls}}$. $x_{\mathrm{cls}}$ will be inserted at the beginning of each input sequence to accumulate information from other tokens. We compute the token score from its attention score $a_{\mathrm{cls}} = \mathtt{softmax}(\frac{q_{\mathrm{cls}}\mathcal{K}^\top}{\sqrt{d}}) \in \mathbb{R}^{\mathrm{L}}$, where $q_{\mathrm{cls}}$ is the query vector of $x_{\mathrm{cls}}$. The attention score comes from the intrinsic attention mechanism in our transformer-based backbone do not require any additional cost. Each element in $a_{\mathrm{cls}}$ corresponds to a token in the input sequence and is expressed as its associated token score.

**Details of Dynamic Patch Selection Mechanism.** We first split all WSIs of each example to construct a patch bank, then, as nailed in Fig. 3, we identify the neighborhood tokens during each item at first. We select $b$ key tokens with Top-$b$ attention scores, usually, the value of $b$ is ding to 4. A more comprehensive discussion on the number of $b$ key tokens can be found in Sec. 4.4, where additional details are provided.

We use ResNet to encode fixed-size image sub-regions, resulting in different image sizes producing varying numbers of tokens. For example, for SKCM, the average number of tokens is 58,381, the maximum is 1,010,257, and the minimum is 923. This variation in the number of tokens makes parallel training challenging, as it requires input data to have the same shape to form a batch for network training. CancerMoE addresses this issue by fixing the number of tokens for each input WSI to ( N ), enabling parallel training. Then extract $\mathrm{N} \times (1-p)$ tokens around these $b$ tokens as part of selected tokens for DPS, where $p$ is the ratio of selected unseen tokens. For the remaining $\mathrm{N} \times p$ tokens of DPS, we uniformly select them among the unused original tokens to explore more informative tokens and avoid overfitting to neighborhood tokens. However, the weight of the model is continuously updated during the training epoch, which actively keeps changing the attention value of the same token. Hence, for a stable output, we update the token score only when the c-index of the training set decreases. For more details about the token score update method and DPS please refer to Section 4.4.

### 3.4 Attention Consolidation and Sparsification

---
**Algorithm 1** Attention Sparsification, $\mathcal{A}_i(x)$

---
**Require:** query, key, and value of $x$ $\mathcal{Q}, \mathcal{K}, \mathcal{V}$
 1: $\mathcal{A} = \mathtt{softmax}(\frac{\mathcal{Q}\mathcal{K}^\top}{\sqrt{d}})$ # Calculate the attention map
 2: Calculate $k \leftarrow (q \times \mathrm{N})^2$, $\mathcal{A}_{\mathrm{flat}} \leftarrow \mathtt{Flatten}(\mathcal{A})$
 3: $\mathcal{A}_{\mathrm{top}} \leftarrow \mathtt{TopK}(\mathcal{A}_{\mathrm{flat}}, k)$, $\mathcal{A}_{\mathrm{sparse}} \leftarrow \mathtt{Reshape}(\mathcal{A}_{\mathrm{top}})$
 4: **return** $\mathcal{A}_{\mathrm{sparse}}\mathcal{V}$

---

Owing to the redundancy among attention heads Michel et al. (2019); Beltagy et al. (2020), it motivates us to consolidate learned information, which can enable efficient prediction of cancer patient's survival rates.

---

**Algorithm 2** Attention Consolidation

---

**Require:** Attention importance scores: $\{\mathtt{I}_1, \mathtt{I}_2, \ldots, \mathtt{I}_\mathcal{H}\}$
**Ensure:** Attention Head output: $\{\mathcal{A}_1(\boldsymbol{x}), \mathcal{A}_2(\boldsymbol{x}), \mathcal{A}_\mathcal{H}(\boldsymbol{x})\}$

1: $\{\mathcal{A}_1^{(1)}, \mathcal{A}_1^{(2)}, \cdots, \mathcal{A}_1^{(k)}\} = \texttt{Topk}(\{\mathtt{I}_1, \mathtt{I}_2, \ldots, \mathtt{I}_\mathcal{H}\}, \{\mathcal{A}_1(\boldsymbol{x}), \mathcal{A}_2(\boldsymbol{x}), \mathcal{A}_\mathcal{H}(\boldsymbol{x})\})$ # Use importance scores to select important heads
2: $\boldsymbol{y} = \{\}$ # Attention output
3: **for** $\mathcal{A}_1^{(i)}$ in $\{\mathcal{A}_1^{(1)}, \mathcal{A}_1^{(2)}, \cdots, \mathcal{A}_1^{(k)}\}$ **do**
4:      $\mathcal{A}_{cluster}^{(i)} = \{\}$ # Attention Cluster
5:      $\mathcal{I}_{cluster}^{(i)} = \{\}$ # Attention importance score in the cluster
6:      **for** $\mathcal{A}_j(\boldsymbol{x})$ in $\{\mathcal{A}_1(\boldsymbol{x}), \mathcal{A}_2(\boldsymbol{x}), \mathcal{A}_\mathcal{H}(\boldsymbol{x})\}$ **do**
7:          **if** $\texttt{COSINE}(\mathcal{A}_1^{(i)}(x), \mathcal{A}_j(\boldsymbol{x})) \leq \{\texttt{COSINE}(\mathcal{A}_1^{(t)}(x), \mathcal{A}_j(\boldsymbol{x}))\}_{t \neq i, t=1}^{t=k}$ **then**
8:              Add $\mathcal{A}_j$ to $\mathcal{A}_{cluster}^{(i)}$
9:              Add $\mathcal{I}_j$ to $\mathcal{I}_{cluster}^{(i)}$
10:          **end if**
11:      **end for**
12:      $y_i \leftarrow \boldsymbol{0}$
13:      **for** $A_i$ in $\mathcal{A}_{cluster}^{(i)}$ **do**
14:          $y_i = y_i + \texttt{softmax}(\mathcal{I}_{\texttt{cluster}}^{(\texttt{i})})_i \times \mathcal{A}_i(\boldsymbol{x})$
15:      **end for**
16:      Add $y_i$ to $\boldsymbol{y}$
17: **end for**

---

Our attention consolidation and sparsification (ACS) algorithm consists of two components: (1) *attention consolidation*, where attention maps are clustered based on their cosine similarity and then merged into a few more knowledge ones; (2) *attention sparsification*, where superfluous attention connections are trimmed for extra inference efficiency.

▷ *Attention Consolidation.* As shown in Fig. 2 and Algorithm 2, we first calculate the importance score $\{\mathtt{I}_1, \mathtt{I}_2, \ldots, \mathtt{I}_\mathcal{H}\}$ of all attention heads $\{\mathcal{A}_i\}|_{i=1}^{\mathcal{H}}$ to identify the most informative ones, where $\mathcal{H}$ is the number of attention heads. To be specific, the importance of attention head $\mathcal{A}_i$ is estimated as:

$$\mathtt{I}_i = \mathbb{E}_{\boldsymbol{x} \sim \mathcal{X}} \left| \mathcal{A}_i(\boldsymbol{x})^\top \frac{\partial \mathcal{L}(\boldsymbol{x})}{\partial \mathcal{A}_i(\boldsymbol{x})} \right|, \tag{2}$$

where $\mathcal{X}$ symbolizes the distribution of training data, $\mathcal{L}(\boldsymbol{x})$ signifies the objective function, and $\mathcal{A}_i(\boldsymbol{x})$ represents the output features.

Then, $k$ informative attention heads are selected, and $\texttt{K-means}$ is applied to assign the rest of the attention heads to these $k$ informative ones, according to their cosine similarities. In this way, $\{\mathcal{A}_1^{(1)}, \mathcal{A}_1^{(2)}, \cdots, \mathcal{A}_1^{(k)}\}$ denotes the $k$ selected attention heads. Their associated sets of clustered heads are $\{\{\mathcal{A}_2^{(1)}, \cdots, \mathcal{A}_{n_1}^{(1)}\}, \{\mathcal{A}_2^{(2)}, \cdots, \mathcal{A}_{n_2}^{(2)}\}, \cdots, \{\mathcal{A}_2^{(k)}, \cdots, \mathcal{A}_{n_k}^{(k)}\}\}$, where $\{n_1 - 1, \cdots, n_k - 1\}$ are the number of allocated heads and $\sum_{i=1}^{k} n_i = \mathcal{H}$. The output $\boldsymbol{y}_i$ from the cluster $i$ is described as a weighted sum across $n_i$ heads:

$$\boldsymbol{y}_i = \overbrace{\texttt{softmax}(\{\mathtt{I}_1, \cdots, \mathtt{I}_{n_i}\})_i \times \mathcal{A}_i(\boldsymbol{x})}^{\text{The } i\texttt{th} \text{ informative attention head}} + \overbrace{\sum_{j=2}^{n_i} \texttt{softmax}(\{\mathtt{I}_1, \cdots, \mathtt{I}_{n_i}\})_j \times \mathcal{A}_j(\boldsymbol{x})}^{\text{Allocated attention heads in the cluster } i}. \tag{3}$$

The final output from the consolidated multi-head attention can be formulated as $\boldsymbol{y} = \texttt{Concat}(\{\boldsymbol{y}_i\}|_{i=1}^{k})$.

▷ *Attention Sparsification.* In Fig. 2 and Algorithm 1, to remove superfluous attention connections, we further sparsify attention maps by only preserving $(q \times \mathrm{N})^2$ attention elements with the largest magnitude. $q$ is a pre-defined ratio for the attention sparsification and N represents the number of WSIs' tokens. Note that,

Table 1: Performance comparison of our model vs. diverse baselines on 12 cancer diagnostic datasets. The notation "P." signifies the utilization of pathological images, "G." indicates the use of genomic profiles, and "M." implies the incorporation of both pathological images and genomic profiles. We mark the best performance in **bold** and the second best performance in underline.

| Method | Modality | BLCA | BRCA | HNSC | KIRC | KIRP | LIHC | LUAD | LUSC | PAAD | SKCM | STAD | UCEC | Overall↑ |
|---|---|---|---|---|---|---|---|---|---|---|---|---|---|---|
| SNN(NeurIPS'17) Klambauer et al. (2017) | G. | 0.632 | 0.573 | 0.577 | 0.665 | 0.707 | 0.570 | 0.591 | 0.522 | 0.537 | 0.519 | 0.545 | 0.601 | 0.596 |
| OmicMlP(Preprint'23) Jaume et al. (2023) | G. | 0.581 | 0.589 | 0.542 | 0.658 | 0.740 | 0.541 | 0.582 | 0.507 | 0.578 | 0.590 | 0.527 | 0.604 | 0.587 |
| AttnMISL(ICML'18) Ilse et al. (2018) | P. | 0.553 | 0.561 | 0.543 | 0.577 | 0.622 | 0.629 | 0.564 | 0.555 | 0.538 | 0.621 | 0.559 | 0.617 | 0.581 |
| DeepAttnMISL(MIA'20) Yao et al. (2020) | P. | 0.596 | **0.681** | 0.569 | 0.508 | 0.698 | 0.625 | **0.647** | 0.558 | 0.594 | 0.632 | 0.567 | **0.743** | 0.618 |
| Patch-GCN(MICCAI'21) Chen et al. (2021a) | P. | 0.560 | 0.580 | 0.562 | 0.524 | 0.644 | 0.671 | 0.585 | 0.571 | 0.585 | 0.666 | 0.541 | 0.629 | 0.611 |
| TransMIL(NeurIPS'21) Shao et al. (2021) | P. | 0.529 | 0.524 | 0.602 | 0.533 | 0.605 | 0.650 | 0.476 | 0.498 | 0.538 | 0.637 | 0.523 | 0.538 | 0.554 |
| MCAT(ICCV'21) Chen et al. (2021b) | M. | 0.624 | 0.580 | 0.557 | 0.661 | 0.771 | 0.636 | 0.620 | 0.503 | 0.627 | 0.613 | 0.514 | 0.622 | 0.610 |
| CLAM-SB(Nat. Biomed. Eng.'21) Lu et al. (2021) | P. | 0.549 | 0.598 | 0.577 | 0.573 | 0.610 | 0.645 | 0.566 | 0.545 | 0.541 | 0.629 | 0.562 | 0.599 | 0.583 |
| CLAM-MB(Nat. Biomed. Eng.'21) Lu et al. (2021) | P. | 0.553 | 0.585 | 0.541 | 0.567 | 0.623 | 0.630 | 0.565 | 0.561 | 0.554 | 0.626 | 0.566 | 0.581 | 0.579 |
| PorpoiseAMIL(Cancer Cell'22) Chen et al. (2022b) | P. | 0.542 | 0.560 | 0.564 | 0.567 | 0.539 | 0.618 | 0.548 | 0.561 | 0.580 | 0.607 | 0.556 | 0.638 | 0.584 |
| MMF(Cancer Cell'22) Chen et al. (2022b) | M. | 0.627 | 0.558 | 0.580 | 0.711 | 0.811 | 0.640 | 0.586 | 0.527 | 0.591 | 0.608 | 0.587 | 0.644 | 0.629 |
| Surformer(CMPB'23) Wang et al. (2023) | P | 0.553 | 0.623 | 0.576 | 0.520 | 0.594 | **0.678** | 0.580 | 0.549 | 0.544 | 0.640 | **0.606** | 0.592 | 0.588 |
| CMTA(ICCV'23) Zhou & Chen (2023) | M | 0.619 | 0.613 | 0.587 | 0.617 | 0.802 | 0.567 | 0.642 | **0.646** | 0.556 | 0.590 | 0.556 | 0.590 | 0.616 |
| Ours | M. | **0.653** | 0.576 | **0.603** | **0.752** | **0.824** | 0.647 | 0.644 | 0.571 | **0.634** | **0.687** | 0.605 | 0.660 | **0.655** |

at the inference phase, the attention calculation purely happens among the selected $q \times N$ tokens, leading to substantially reduced computational costs. Finally, these consolidated and sparsified tokens are processed by task-specific heads to determine the cancer prognosis.

## 3.5 Multi-modal Fusion for Dynamic Patch Selection

With the DPS and ACS, we have achieved substantial training efficiency. Nonetheless, the performance remains suboptimal. We hypothesize that the limitation arises from the inadequacy of token selection by the DPS. To address this issue, integrating additional modalities is proposed to further enhance the DPS and then achieve more competitive performance. We pack adjacent genes into tokens to construct the genomic sequence as an additional modality. In our `CancerMoE` framework, we consolidate all modalities pertaining to a single input into a unified sequence. This is achieved by leveraging the self-attention mechanism to fuse cross-modal information. This design strategy not only facilitates the seamless integration of multi-modal data without necessitating structural modifications but also promotes the DPS by effectively utilizing other modal information. The benefits and advancements of our design are further elaborated in Table 3.

# 4 Experiment

## 4.1 Implementation Details.

**Datasets.** To evaluate our proposed `CancerMoE`, we conduct experiments on The Cancer Genome Atlas (TCGA), a publicly accessible database housing genomic and clinical data derived from thousands of cancer patients, encompassing 33 prevalent cancer types. The Cancer Genome Atlas (TCGA) is a publicly accessible database housing genomic and clinical data from thousands of cancer patients, encompassing 33 prevalent cancer types commonly used in cancer prognosis prediction. We select 12 cancer types that are frequently used in plenty of works Chen et al. (2021c); Klambauer et al. (2017); Jaume et al. (2023); Ilse et al. (2018); Chen et al. (2021a); Shao et al. (2021; 2023); Chen et al. (2021b); Lu et al. (2021); Chen et al. (2022b). We utilize the pre-processed Whole Slide Image (WSI) is proposed by Chen et al. (2022b) as the image input for `CancerMoE`. The WSI is segmented $256 \times 256$ sub-images of high-resolution histology images and extracts each sub-image into feature vector $\mathbb{R}^{1024}$ by CLAM Lu et al. (2021). The size of the different WSIs varies greatly ($8,417 \times 6,602$ to $191,352 \times 91,562$), resulting in the number of paths $\mathbb{R}^{1024}$ also varies accordingly, which makes parallelizing the training process difficult. More details about datasets and implementation can be found in Appendix A.

**Optimization Object.** To optimize the model parameters, we utilize the log-likelihood function for a discrete survival model Chen et al. (2022b), $L_c$, where $L_c$ is the loss function for the censor patients. Formally, the survival state of a patient considers two factors: 1) Censorship status, where $c = 0$ signifies an observed patient death and $c = 1$ corresponds to the patient's last known follow-up. 2) Time-to-event, represented as $t_i$, signifies the duration between the patient's diagnosis and observed death if $c = 0$, or the time until the

last follow-up if $c = 1$. The $h$ denotes the output representing discrete survival predictions:

$$\texttt{hazards} = \sigma(h). \tag{4}$$

In the next step, the cumulative survival function $S(t)$ is calculated from the hazards:

$$\mathcal{S}(t) = \prod_{i=0}^{t}(1 - \texttt{hazards}_i). \tag{5}$$

Then the final loss function $\mathcal{L}_{\texttt{c}}$ corresponding to censored patients is defined as:

$$\mathcal{L}_{\texttt{c}} = -(1 - c) \cdot (\log(\mathcal{S}(t-1) + \log(\mathcal{S}(t)))). \tag{6}$$

The term of the loss function corresponding to uncensored patients $\mathcal{L}_{\texttt{u}}$ is defined as:

$$\mathcal{L}_{\texttt{u}} = -c \cdot \log(\mathcal{S}(t)). \tag{7}$$

The final loss function can be obtained by combining $\mathcal{L}_{\texttt{c}}$ and $\mathcal{L}_{\texttt{u}}$. The $\beta$ is the hyperparameter that balances the two loss terms.

$$\mathcal{L}_{\texttt{survival}} = (1 - \beta) \cdot L_{\texttt{c}} + \beta \cdot L_{\texttt{u}} \tag{8}$$

**Evaluation Metric.** The performance of the models is assessed using the concordance index (c-index) Harrell et al. (1982), where higher values indicate better performance. The c-index measures the proportion of all possible pairs of observations for which the model's predicted values accurately predict the ordering of actual survival. It ranges from 0.5 (indicating random prediction) to 1 (reflecting perfect prediction). The c-index can be expressed with the following formulation:

$$\texttt{c-index} = \frac{1}{n(n-1)} \sum_{i=1}^{n} \sum_{j=1}^{n} I(\text{T}_i < \text{T}_j)(1 - c_j), \tag{9}$$

where $n$ is the sample size, $\text{T}_i$ and $\text{T}_j$ represents the survival time of the $i$-th and $j$-th patients. The symbol $I(\cdot)$ denotes the indicator function, which evaluates to 1 if its argument is true and 0 otherwise. Meanwhile, $c_j$ indicates the correct censorship status.

We follow the evaluation of Chen et al. (2021c); Klambauer et al. (2017); Jaume et al. (2023); Ilse et al. (2018), which utilize 5-fold cross-validation to demonstrate the superiority of our method.

**Model Implementation Details.** SMoE: We employ two transformer encoder layers, and the SMoE is tailored in the MLP layer of the last transformer encoder layers. The number of experts is 4 or 8, and we use the load and importance balancing loss Shazeer et al. (2017b) to combat the imbalance loading phenomenon Chi et al. (2022). DPS: We use the attention score of the last transformer encoder layers as the token scores. For the neighborhood tokens, assuming the number of neighborhood regions is $N_n$, the total number of tokens is $N$, and the ratio of select unseen tokens is $p$. We will select $N \times (1 - p)/(2N_n)$ tokens on the right and left sides of the Key Token, respectively. ACS: We do consolidation on each transformer encoder layer. The sparsification is only executed in the first transformer encoder layer, where we filter 92% WSIs tokens. Model Architecture: The number of attention heads is 8, the hidden dimension of our model is 32. For the genomic profiles, we use a patch embedding layer that splits each gene profile vector into sequences with length 8. Training: The training batch size is set to 32, and the learning rate is $1e^{-3}$. For other important hyperparameters, we use the same default settings for all cancer types except BRCA, LUSC, and SKCM: 3072 selected tokens, 4 key tokens, and a ratio of 0.5 for selecting unseen tokens.

## 4.2 Powerful performance of `CancerMoE` in fierce competitions

In this section, we fairly compare the performance of our model with various state-of-the-art baselines. The involved machine learning models are SNN Klambauer et al. (2017), OmicMlP Jaume et al. (2023), AttnMISL Ilse et al. (2018), Patch-GCN Chen et al. (2021a), TransMIL Shao et al. (2021), MCAT Chen

Table 2: Parameters, FLOPs, VRAM consumption, and Training time of `CancerMoE` *v.s.* diverse baselines that involve pathological images. The VRAM consumption of each method is in the training stage (Average VRAM consumption across all cancer datasets), and the training time is the average time for all 12 cancers. We mark the best performance in **bold** and the second in underline.

| Method | Modality | Params(M)↓ | FLOPs(G)↓ | VRAM(G)↓ | Training time(H)↓ |
|---|---|---|---|---|---|
| AttnMISL Ilse et al. (2018) | P. | 0.920 | 42.189 | 7.320 | 4.861 |
| Patch-GCN Chen et al. (2021a) | P. | 1.187 | 2.545 | 20.843 | 4.974 |
| TransMIL Shao et al. (2021) | P. | **0.275** | 11.743 | 12.117 | 8.001 |
| MCAT Chen et al. (2021b) | M. | 3.210 | 7.823 | 6.003 | 6.479 |
| CLAM-SB Lu et al. (2021) | P. | 0.790 | 14.707 | 7.007 | 4.327 |
| CLAM-MB Lu et al. (2021) | P. | 0.791 | 39.842 | 8.053 | 6.317 |
| PorpoiseAMIL Chen et al. (2022b) | P. | 0.937 | 40.872 | 13.294 | 5.747 |
| DeepAttnMISL Yao et al. (2020) | P. | 8.532 | 33.294 | 4.417 | 12.047 |
| Surformer Wang et al. (2023) | P. | 14.520 | 18.534 | 4.898 | 4.343 |
| MMF Chen et al. (2022b) | M. | 6.849 | 137.24 | 12.376 | 7.324 |
| Ours | M. | 0.446 | **0.170** | **1.875** | **2.362** |

et al. (2021b) and CMTA(ICCV'23) Zhou & Chen (2023), and biology literature-based methods include CLAM-SB Lu et al. (2021), CLAM-MBLu et al. (2021), MMF Chen et al. (2022b), PorpoiseAMIL Chen et al. (2022b), and Surformer Wang et al. (2023). Given that single-modal approaches still exhibit superior performance for certain cancers, we also compare our model with single-modal baselines that utilize either only pathological images or genomic profiles. The whole comparison results of `CancerMoE` *v.s.* baselines on 12 type of cancers are shown in Table 1, in which we make the following three observations. ① `CancerMoE` achieves the highest overall performance across all 12 cancer datasets. Specifically, `CancerMoE` exceeds biology-based and learning-based baselines {0.026, 0.071, 0.068}, and {0.094, 0.072, 0.062} in cancers {`KIRC`, `BLCA`, `LUAD`}, respectively. These empirical results demonstrate the effectiveness of our model in addressing the cross-modality conflict and assigning plausible SMoE experts to conduct better cancer prognosis prediction. ② On `LIHC` and `BRCA`, the performance of `CancerMoE` merely achieves a moderate level. The best performance of these two cancer types is achieved by methods Patch-GCN Chen et al. (2021a) and CLAM-SB Lu et al. (2021) that only use WSI images, which indicates we need a more comprehensive fusion mechanism to effectively integrate genomic profiling with histological image in LIHC and BRCA. ③ Our multimodal approach outshines competing baselines in resolving modality conflicts across diverse cancer datasets, evident in consistently superior performance metrics in the c-index. By seamlessly integrating information from pathological images and genomic profiles, our model excels in {`KIRC`, `BLCA`, `LUAD SKCM`}, surpassing MMF Chen et al. (2022b) {0.041, 0.021, 0.024, 0.021}, beating MCAT Chen et al. (2021b) {0.091, 0.029, 0.024, 0.074}. These results prove our model's efficacy in leveraging complementary modalities, effectively addressing and reconciling conflicts for enhanced cancer diagnostic accuracy.

### 4.3 Superior Efficiency Across Diverse Baselines.

Given the extremely high dimensionality of image data in pan-cancer diagnosis, we investigate the efficiency of `CancerMoE` compared to baseline models. In Table 2, we advance deeply to demonstrate the advance of `CancerMoE` on efficient training and inference. `CancerMoE` achieve improved performance with much fewer computational resources in terms of fewer data patches and training epochs. The flops of `CancerMoE` is solely **1/1000** of that of MMF Chen et al. (2022b), yet manifests a considerable qualitative improvement. Compared to MCAT Chen et al. (2021b), `CancerMoE` use 1/50 computation complexity, with a 7.4% higher c-index, which clearly shows the superiority and viability of our method. What is even more noteworthy is that with the same granularity choices including batch and patch size, `CancerMoE` only utilizes 9%-20% GPU memory (VRAM) of previous methods. Moreover, in a direct comparison with baselines, `CancerMoE` consistently outperforms in the competition.

### 4.4 Ablation and Additional Investigations.

**Ablation on each component in `CancerMoE`.** To validate the effectiveness of each component in `CancerMoE`, we conduct ablation studies as recorded in Table 3. Results indicate that (1) `ACS` is the central performance contributor; (2) The designs of `DPS` and `SMoE` bring similar level amounts of performance improvements; (3) The combination of above three leads to a superior result in cancer prognosis; (4) The superior performance compared with "$w/o$ `DPS`" that selects tokens randomly, demonstrate the efficacy of `DPS` in finding important tokens. (5) In the `LUSC` dataset, employing genomic profiles without the `SMoE` framework yields inferior results compared to using only WSIs ("$w/o$ `SMoE`" versus "$w/o$ Genomic Profiles"), which suggests the presence of gradient conflicts, in which `SMoE` effectively mitigates. (6) The enhancement in performance when moving from "$w/o$ Genomic Profiles" and "$w/o$ `DPS`" to the `CancerMoE` model demonstrates the benefit of incorporating additional modalities. It leads to selecting elite tokens better by `DPS` and leverages genomics data to promote prediction accuracy.

Table 3: Ablation studies on `DPS`, `ACS`, `SMoE`, and fusion. We fix a set of randomly selected tokens during training to replace the `DPS` as "$w/o$ `DPS`", use the vanilla attention module to replace the `ACS` as "$w/o$ `ACS`", use the dense MLP module with the same parameter to replace the `SMoE` layer as "$w/o$ `SMoE`", and remove genomic profiles as "$w/o$ Genomic Profiles". The "Random Select" replaces the `DPS` policy with policy that keep random token selection during training and inference.

| Setting | BRCA | LUSC | SKCM |
|---|---|---|---|
| `CancerMoE` | **0.576** | **0.571** | **0.687** |
| - $w/o$ `DPS` | 0.564 | 0.519 | 0.655 |
| - $w/o$ `ACS` | 0.541 | 0.506 | 0.641 |
| - $w/o$ `SMoE` | 0.565 | 0.493 | 0.665 |
| - $w/o$ Genomic Profiles | 0.539 | 0.525 | 0.515 |
| - Random Select | 0.555 | 0.529 | 0.567 |

Table 4: Ablation on # selected tokens (N) of `CancerMoE`.

| N | BRCA | LUSC | SKCM |
|---|---|---|---|
| 512 | 0.569 | **0.571** | 0.654 |
| 1024 | **0.576** | 0.547 | 0.651 |
| 2048 | 0.566 | 0.536 | 0.655 |
| 3072 | 0.515 | 0.509 | **0.687** |
| 4096 | 0.546 | 0.536 | 0.643 |

Table 5: Ablation studies on # Key Tokens ($b$) of `CancerMoE`.

| Setting | BRCA | LUSC | SKCM |
|---|---|---|---|
| 1 | 0.570 | 0.548 | 0.626 |
| 2 | 0.573 | 0.527 | 0.636 |
| 3 | 0.570 | 0.559 | 0.627 |
| 4 | 0.571 | **0.571** | **0.687** |
| 5 | **0.576** | 0.567 | 0.642 |

Table 6: Ablation on # informative attention heads in `ACS`.

| Setting | BRCA | LUSC | SKCM |
|---|---|---|---|
| 0.1 | 0.537 | 0.555 | 0.600 |
| 0.3 | 0.561 | 0.544 | **0.687** |
| 0.5 | **0.576** | **0.571** | 0.666 |
| 0.7 | 0.572 | 0.527 | 0.624 |
| 0.9 | 0.571 | 0.565 | 0.655 |

**`DPS` - The Number of Selected Tokens.** The results in Table 4 show that ① The best number of selected tokens is dataset-dependent. Results vary from dataset to dataset. We present clear indications on `BRCA`, `LUSC`, and `SKCM` datasets. For `BRCA`, the performance pinnacle is reached at $N = 1024$, with `LUSC` and `SKCM` arriving at a sweet point for superior predictions at $N$ values of 512 and 3072, correspondingly. ② The performance shows an upward trend as the value of $N$ increases. This observation highlights that too small a number of tokens do not provide enough feature information for the `DPS` to capture. ③ Following its peak, we obviously note that the performance experiences a gradual decline as $N$ values increase, which indicates that abundant tokens do not necessarily yield superior outcomes. Although `DPS` selects quality patches for training, more tokens inevitably introduce noise, affecting performance.

Table 7: Ablation studies on different token score update policies of our proposed `CancerMoE`. "per epoch" denotes the update token score every epoch, "$n$ epoch apart" denotes the update token score $n$ epoch apart, and "c-index depends" denotes token score is updated when the c-index of the training set decrease.

| Setting | BRCA | LUSC | SKCM |
|---|---|---|---|
| per epoch | 0.570 | 0.528 | 0.656 |
| 1 epoch apart | 0.560 | 0.520 | 0.662 |
| 2 epochs apart | 0.564 | 0.523 | 0.652 |
| 4 epochs apart | 0.562 | 0.521 | 0.664 |
| c-index depends | **0.576** | **0.571** | **0.687** |

**`DPS` - The Number of Key Tokens $b$.** The ablation experiments on the number of Key Tokens $b$ are presented in Table 5, it is verified on datasets `BRCA`, `LUSC`, and `SKCM`. The optimal diagnostic benefit is achieved when the value of $b$ is set to 4, too small or too large of $b$, both causing performance degradation. The finding shows the importance of the number of neighborhood tokens, which is crucial to identifying diagnostic information.

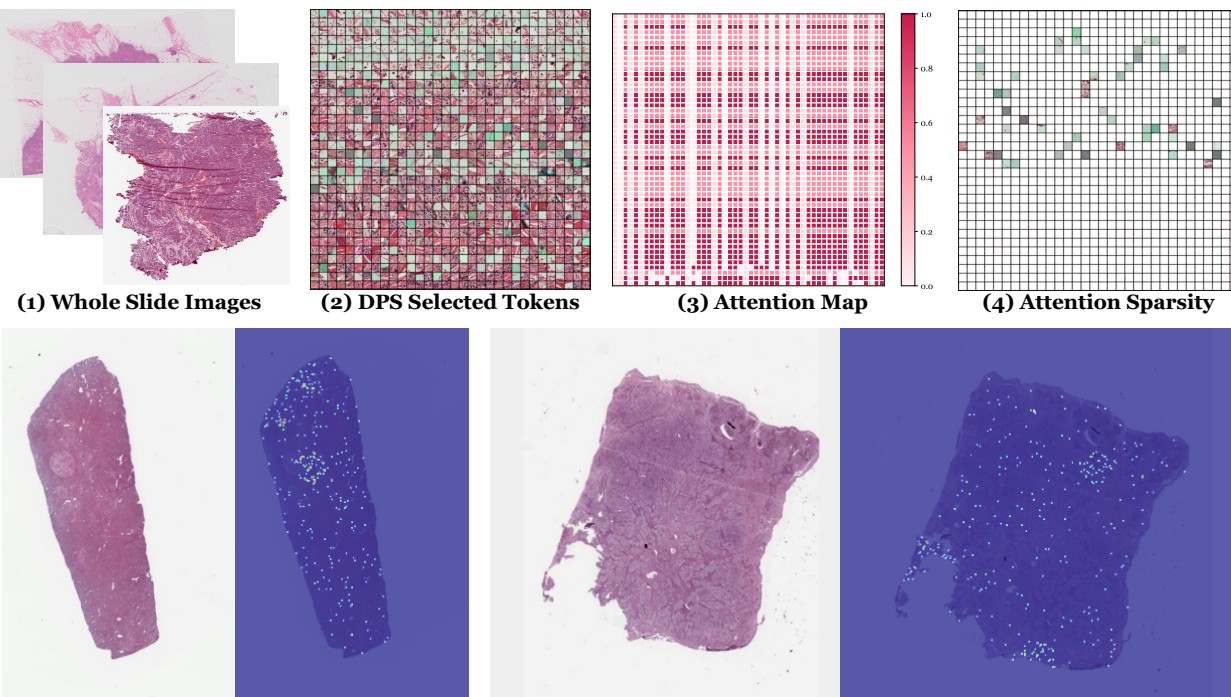

Figure 4: **Top**: Analysis of the diversified attention via attention sparsity. These blocks are selected tokens via critical region identification, and colorful blocks are unmasked attention tokens. **Bottom**: Additional visualized attention scores identified by CancerMoE.

**DPS - Token Score Update Policy.** As shown in Table 7, adopting different token score update policies influences the performance of `CancerMoE`. On the `BRCA`, `SKCM` cancer dataset, the "c-index depends" approach exhibits superior predictive capabilities, outperforming the "4 epochs apart" and "2 epochs apart" strategies. Moreover, the "c-index depends" tactics demonstrate more impressive competitiveness on the `LUSC` dataset. Compared with the sub-optimal one, the "c-index depends" exceeds 0.043. Considered holistically, we determine that "c-index depends" represents the most effective token score update policy.

**DPS - The Ratio $p$ of Select Unseen Tokens.** The ratio $p$ of select unseen tokens indicates how much the overview information we use for prognosis prediction. We also conducted extensive investigations on the ratio $p$ illustrated in Table 6. Initially, as the ratio increased, the accuracy of prognostic diagnosis improved. Subsequently, the optimal performance plateaued at $p = 0.5$, reaching a saturation state. Increasing the parameter $p$ allows the model to encounter a broader range of new tokens, thereby mitigating the risk of overfitting to a limited set of specific tokens. However, setting $p$ too high can be counterproductive, as it may lead the model to sample tokens too randomly, which can obstruct the model's ability to converge effectively. The observation highlights that the balance between local and overview WSI information is critical and needs to be carefully determined.

**ACS - Consolidation and Sparsification** In order to substantiate our proposition that eliminating redundant information carried by redundant attention heads can result in remarkable advancements in cancer prognostic performance, we performed fusion experiments by varying the diverse number of attention heads, and the resultant findings are presented in Table 8. The data reveals that aggregating multiple heads brings substantial advantages without any accompanying disadvantages. Notably, the most gratifying outcomes are obtained when 2 is chosen as the number of informative attention heads.

Table 8: Ablation studies on the ratio $p$ of selected unseen tokens of our proposed `CancerMoE`.

| Setting | BRCA | LUSC | SKCM |
|---|---|---|---|
| *w/o Consolidation* | 0.536 | 0.541 | 0.685 |
| 1 | 0.540 | 0.511 | 0.682 |
| 2 | **0.576** | 0.514 | **0.687** |
| 3 | 0.567 | **0.571** | 0.671 |

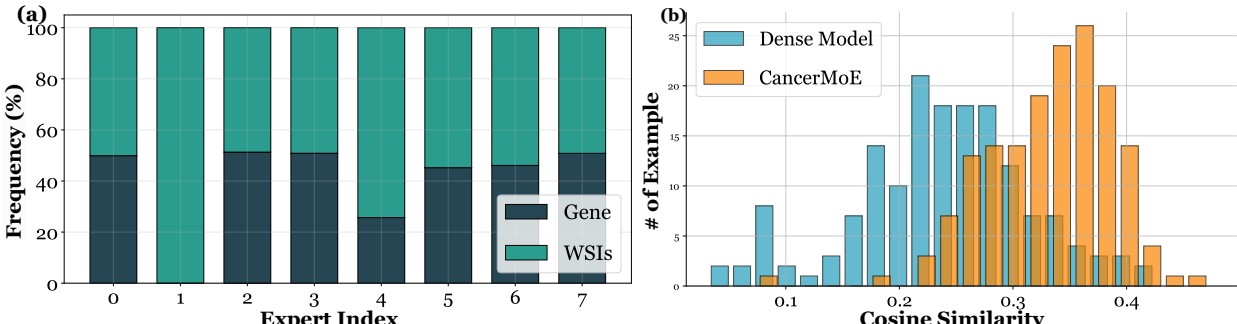

Figure 5: **(a)** The expert selection across different modalities in the `BRCA` dataset. **(**b) The gradient conflict between modalities in `CancerMoE` and the dense counterpart. Here, the gradients are obtained from the experts and dense MLP with the same configuration in `CancerMoE` and Dense Model, respectively. our proposed sparse model demonstrates reduced conflict, as evidenced by more positive cosine distances, thereby facilitating enhanced multi-modal integration. The `BRCA` dataset is used for the experiment. The gradient is collected from the last transformer layer. More positive cosine distances denote less gradient conflict.

**`ACS` - Interpretability from Diversified Attention.** To raise the interpretability of the model, we conducted experiments with a sparse algorithm for the self-attention module. The outcomes of the sparsity operation are displayed in Fig. 4, where we eliminate elements with low information content in the sequence to reduce inherent redundancy and improve efficiency.

**`SMoE` - Modality Level Routing Specialization.** To showcase the effectiveness of the modality router structure, we present a visualization of `CancerMoE` in Fig. 5 **(a)**. It can be observed that the expert 1 and the expert 4, who ponder to be attributed to genomic profiles and others, tend to process both modalities.

**Gradient Conflict between Modalities.** As previously mentioned, our modality-specific routing policy directs modality embeddings towards compatibility experts, which in turn produce high-quality modality features. This strategy effectively addresses various modalities and segregates the network parameter space according to different modalities and tasks. As demonstrated in Fig. 5 **(b)**, disentangling the model's parameter space significantly reduces gradient conflict between modalities. This separation leads to enhanced performance, which is further demonstrated in Table 3.

## 5 Conclusion and Limitation

This paper proposes `CancerMoE`, a multi-modal cancer prognosis prediction pipeline, to address the high computing costs incurred by WSIs and the gradient conflict arising from the heterogeneity between histological and genomic data. Firstly, in `CancerMoE`, the Dynamic Patch Selection (DPS) module tackles the complexity of ultra-high resolution by only feeding elite patches. Then, the Sparse Mixture-of-Experts (SMoE) is tailored to disentangle model parameter space to mitigate the gradient conflict. Finally, the Attention Consolidation and Sparsification (ACS) mechanism is investigated to diminish attention redundancy and enhance the efficiency of training and inference steps. Our `CancerMoE` has demonstrated superior performance on cancer prognosis prediction, with the c-index significantly increasing in 12 types of cancer and beating all other methods. Moreover, the experiments indicate that `CancerMoE` is more efficient than SoTA methods in terms of FLOPs, VRAM, and training time. Our approach offers valuable insights and techniques for multimodal AI to aid in efficient cancer prognosis. This fosters interdisciplinary progress across biology, medicine, and computer science. As medical AI rapidly evolves, using multimodal AI in cancer prognosis is becoming a practical reality.

`CancerMoE` has exhibited its effectiveness and exceptional performance in cancer prognosis tasks through experiments on multiple cancer datasets; nevertheless, apart from histopathology images and genomics, there exist multiple other modalities, such as EHR (Electronic Health Records). Our future vision entails the establishment of a multi-cancer types medical diagnostic service, incorporating these diverse modalities

to enhance the capabilities of our proposed approach. The integration of additional modalities into our framework poses an intriguing question that necessitates further exploration.

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

This is the supplementary material for One Leaf Knows Autumn: A Piece of Data-Model Facilitates Efficient Cancer Prognosis with Histological and Genomic Modalities.

We provide the following materials in this manuscript:

- Sec. A the details of datasets and baselines.

- Sec. B the title explanation.

## A    Datasets and Baseline Details

**Datasets Details**    We have utilized data from 12 public cancer types sourced from The Cancer Genome Atlas (`TCGA`) Program for our experiments. These cancer types include Bladder Urothelial Carcinoma (`BLCA`), Breast Invasive Carcinoma (`BRCA`), Head and Neck Squamous Cell Carcinoma (`HNSC`), Kidney Renal Clear Cell Carcinoma (`KIRC`), Kidney Renal Papillary Cell Carcinoma (`KIRP`), Liver Hepatocellular Carcinoma (`LIHC`), Lung Adenocarcinoma (`LUAD`), Lung Squamous Cell Carcinoma (`LUSC`), Pancreatic Adenocarcinoma (`PAAD`), Skin Cutaneous Melanoma (`SKCM`), Stomach Adenocarcinoma (`STAD`), and Uterine Corpus Endometrial Carcinoma (`UCEC`), totally involving hundreds of patients and Hematoxylin and Eosin (H&E) diagnostic Whole Slide Images (WSIs). The elaborate information regarding these datasets is provided in Table 9. Thousands of genomic features are compiled for each patient, sourced from Copy Number Variation (CNV) data, mutation status, and bulk RNA-Seq expression derived from the differentially expressed genes. This data is collected from The Cancer Genome Atlas (TCGA) and the cBioPortal Cerami et al. (2012).

**Baseline Details.**    To facilitate a thorough comparison, we implement and assess various survival prediction methods using the same 5-fold cross-validation splits. These methods encompass both the single-modal learning paradigm and the multi-modal learning paradigm. The experimental results for all these methods across the 12 TCGA datasets are summarized in Table 1. The Params and FLOPs of `CancerMoE` and all baseline methods are calculated on the `BRCA` dataset. For feature extraction, once segmentation is completed, image patches of dimensions $256 \times 256$ are extracted without overlapping, based on the $20\times$ equivalent pyramid level from all identified tissue regions. Following this, a pre-trained ResNet50 model, which had been trained on Imagenet, is employed as an encoder. It converted each $256 \times 256$ patch into a 1024-dimensional feature vector using spatial average pooling after the third residual block.

**Baseline Modal.**    Machine learning models: 1) SNN Klambauer et al. (2017): It is a self-normalizing network model, which serves as the single-modal baseline when working exclusively with genomic profiles. 2) OmicMLP Haykin (1998); Jaume et al. (2023): It utilizes a 4-layer Multi-Layer Perceptron (MLP). 3) AttnMISL Ilse et al. (2018): It employs gated-attention pooling for the WSIs. 4) Patch-GCN Chen et al. (2021a): It explores a hierarchical aggregation approach to consolidate image-level features. 5) TransMIL Shao et al. (2021): TransMIL approximates patch self-attention using the Nyström method Xiong et al. (2021).6) MCAT Chen et al. (2021b): MCAT employs Genomic-Guided Co-Attention (GCA), a mechanism similar to the standard transformer attention that serves the purpose of establishing relationships between image-grid data and word embeddings, much like in the context of VQA Vaswani et al. (2017). Biology literature-based methods: 1) PorpoiseAMIL Chen et al. (2022b): It is mainly based on the attention module, projection, and prediction layers. 2) CLAM-SB and CLAM-MB Lu et al. (2021): After segmentation of WSIs using Clustering-constrained Attention Multiple (CLAM) instance learning's method Lu et al. (2021), survival prediction is performed in two ways. 3) MMF Chen et al. (2022b): An approach is taken to incorporate a multimodal fusion layer, an extension of Pathomic Fusion Chen et al. (2020), to merge the features from SNN and PorpoiseAMIL.

## B    Title Explanation

**One Leaf Knows Autumn**    Generally, the passage of time can be likened to the shifting seasons of the year, each akin to a chapter in the story of one's life. For cancer patients, their journey can often mirror the autumn season, a stage marked by the undeniable symbols of harvest and beauty but one that quickly gives way to the withered chill of winter. In the face of this challenging transition, our mission is to offer patients a

Table 9: TCGA 12 Cancers Case number and Feature Summary.

| Cancer | WSIs | Genomics Profile |
|--------|------|------------------|
| BLCA | 437 | 20404 |
| BRCA | 1021 | 20980 |
| HNSC | 437 | 2217 |
| KIRC | 350 | 2513 |
| KIRP | 284 | 1587 |
| LIHC | 346 | 2583 |
| LUAD | 515 | 21155 |
| LUSC | 484 | 2416 |
| PAAD | 180 | 1659 |
| SKCM | 268 | 2350 |
| STAD | 372 | 2543 |
| UCEC | 539 | 9081 |

glimpse of tackling the impending autumn through a small leaf to help them prepare for the seasons ahead. We aim to usher in the early awareness of autumn in the lives of cancer patients. By equipping with our methods, we can provide better support during the autumn phase of their prognosis, extending the chapters of happiness in their lives. This is the meaning of One Leaf Knows Autumn: to use the most critical parts of histopathology images and genetic signatures to help cancer patients with better treatment and prognosis.

