# OpenReview forum: "One Leaf Knows Autumn: A Piece of Data-Model Facilitates Efficient Cancer Prognosis with Histological and Genomic Modalities"
_TMLR — Rejected by TMLR_

### Review · Reviewer_wixR · 2024-09-24

**Summary Of Contributions:**

To address computing bottlenecks in cancer prognosis analysis, this paper proposed CancerMoE including Dynamic Patch Selector (DPS) and Attention Consolidation and Sparsification (ACS). The DPS dynamically scores and selects the most crucial image patches, minimizing memory overhead by processing only a small fraction of the total image patches. Meanwhile, ACS reduces intrinsic redundancy and enhances efficiency by merging insignificant attention heads into a few informative ones, and pruning less connections within their attention maps. By leveraging a reduced number of image patches (e.g., 1% of total patches) and model parameters (approximately 20%), CancerMoE achieved state-of-the-art performance across twelve cancer datasets, surpassing existing methods in both accuracy and efficiency. Despite these promising results, some limitations still need to be addressed for a more comprehensive optimization.

**Audience:**

Yes

**Broader Impact Concerns:**

The large-size pathology slides are difficult to process at scale, thus developing efficient compute solution is of interest to the community. To assess the performance, authors proposed the CancerMoE for sampling image patches that leads to improve efficient analysis performance on the patient prognosis task. Solid comparison is further needed to justify the real value of the framework for a broad use.

**Claims And Evidence:**

Yes

**Requested Changes:**

1) In Section 3.2，authors claim that “The DPS retrieves b tokens with the highest token scores”. However, in the experiment, b is set to 4. It remains a very small value, especially when considering that the patches typically number in the thousands. Even after incorporating neighboring and random patches, the selection remains limited. Please further discuss why strong performance can be achieved using only 1% of the patch.

2) Random sampling is a common approach in whole slide image. It involves selecting patches randomly from the entire image, and repeated random sampling during model training can effectively cover a broad range of patches. To further validate the effectiveness of DPS, an experiment comparing DPS with random sampling under various patch numbers (e.g., 1%, 3%, 5%) should be conducted.

3) In the ablation study titled "The Number of Selected Tokens," the authors present results for scenarios where the number of selected tokens is set to 512, 1024, or 2048. However, this approach raises some confusion, as ablation studies typically focus on varying the feature dimension rather than adjusting the number of tokens. The token count is generally fixed based on the patch and window sizes in most standard approaches.

4) In Section 3.3, the authors described several operations that are crucial to the functioning of the proposed ACS. However, without a detailed guidance, understanding the exact implementation of these operations is quite challenging for readers. To improve readability and comprehension, it is highly recommended to include pseudocode that outlines these operations step by step.

5) Comparison methods for integrating pathology images and genomic profiles are indeed limited in the current study. To provide a more comprehensive comparison, it is recommended to refer to the method presented in CMTA [1], which offers a relevant approach for multi-modal analysis involving both pathology images and genomic data.

[1] Zhou F, Chen H. Cross-modal translation and alignment for survival analysis[C]//Proceedings of the IEEE/CVF International Conference on Computer Vision. 2023: 21485-21494.

**Strengths And Weaknesses:**

Strength:
1 A good contribution on the efficiency improvement using the proposed approach for pathology data.
2 Experimental results were positive on multiple data sets and cancer types.

Weakness:
1 Notable places have clarity issues that required further improvement.
2 Patch sampling is an important factor to measure the robustness and validity of the work, additional experiments still were needed to justify the use of chosen image patches.
3 A limited comparison with strong baselines.

---

> ### Author Response · Authors · 2024-11-20
> **Response to Reviewer wixR**
>
> **<Cons1> Why DPS improve the performance.**
>
> We begin by noting that b represents the number of key tokens. After obtaining these b tokens, DPS selects adjacent tokens centered around each key token. As mentioned in Section 3.2, we choose (\frac{N \times p}{b}) tokens for each key token. Our superior performance compared to baselines in Table 1, and ablation studies shown in Tables 3 (CancerMoE v.s. “- w/o DPS”) and Table 5, demonstrates that: (1) effectively selecting a small number of tokens with valuable information is good enough to improve survival prediction, and (2) just 4 key tokens are sufficient to identify informative regions for our survival prediction.
>
> **<Cons2> Compare with random sampling and use the percentage of image patch (e.g., 1%, 3%, 5%)**
>
> In Table 3, we should emphasize that we have already compared our method with the random sampling strategy. As noted, the "w/o DPS" approach uses a fixed set of randomly sampled tokens for training. To further address the reviewers' concerns, we incorporated randomly sampled patches and repeated random sampling during model training. Specifically, we conducted an ablation study that replaced the DPS policy with this random sampling method. The results in Table 3 demonstrate that DPS still delivers the best performance, indicating that DPS effectively identifies meaningful regions for survival prediction.
>
> We use a fixed number of tokens rather than a percentage of image tokens for two reasons: (1) to ensure a fair comparison with CancerMoE, and (2) because the number of patches varies across instances. For example, in SKCM, the average, maximum, and minimum number of tokens are $58,381$, $1,010,257$, and $923$, respectively. Our training pipeline does not support inputting a percentage of image tokens into the network for training.
>
> Using a fixed number of tokens allows us to construct consistent batches for training, which improves efficiency (Table 2, the efficient comparisons between CancerMoE and baselines). Despite methods like MMF, which uses varying numbers of tokens for training, our CancerMoE still achieves superior performance (Table 1).
>
> **<Cons3> The Illustration about the “The Number of Selected Tokens”**
>
> In our ablation study, we aim to identify key hyperparameters that could affect prediction performance. The number of selected tokens and the number of key tokens are additional hyperparameters introduced by DPS, so it's important to investigate their impact on the final performance. Although the number of feature dimensions is also an important hyperparameter, it is not directly related to our proposed methods. In our application, the token count indicates how large region sizes are observed for the final survival prediction. Therefore, we use different numbers of tokens to examine how the size of the observed regions affects the final performance.
>
> **<Cons4> The pseudocode of ACS.**
>
> Thanks point it out, this is definitely a very helpful suggestion. We include the pseudocode in our revision (Algorithm 1 and Algorithm 2).
>
> **<Cons5> More multimodal baseline**
>
> Thank you for your valuable suggestion of an advanced relevant baseline. We have included CMTA as an additional multimodal baseline for a more comprehensive comparison. As shown in Table 1 and the subsequent analysis, CMTA outperforms CancerMoE in LUSC and BRCA. However, CancerMoE still achieves the best average performance overall.

---

### Review · Reviewer_R9KF · 2024-10-15

**Summary Of Contributions:**

The paper introduces CancerMoE, a multi-modal framework designed to improve cancer prognosis by integrating histological and genomic data. The authors address significant challenges in computational pathology, such as memory overheads, computing costs, and gradient conflicts, using innovative techniques like dynamic patch selection and a Sparse Mixture-of-Experts (SMoE) framework.

**Audience:**

Yes

**Claims And Evidence:**

Yes

**Requested Changes:**

1. One of the contributions of the paper is Dynamic Patch Selector (DFS) to select a few patches to reduce the massive memory and computing costs. However, in section 3.2, the paper claimed: “tokens processed by the network will be assigned a token score, hereafter referred to as Selected Tokens.”. I wonder if this means that DFS needs a separate neural network to assign a score to each token, but this goes against the purpose of the article.

2. In Figure 2, we can see that the proposed model uses pathological images and molecular profiles. Still, there is little introduction to how to process molecular profiles: what encoder is used, GNN or something else? How to fuse the features of two modalities? It is recommended to provide a more detailed introduction.

3. The ablation experiment in Table 3 shows that the performance improvement mostly comes from multi-modality fusion with genomic profiles. However, in Table.1, most of the comparison methods are not multi-modality methods, which is not fair. Besides, referring to the result of BRAC without genomic profiles in Table 3, the proposed method achieves 0.539, which is lower than most of the methods with only pathological images in Table 1.

4. The paper is not well organized and written. There are some typos, such as formula (9): mathrm. The (3) Attention Map of fig. 4 is not clear.

**Strengths And Weaknesses:**

Strengths:

Innovative Approach:
The use of dynamic patch selection to reduce memory usage is a novel contribution, effectively addressing the bottleneck caused by high-resolution histology images.
Efficiency Improvements:
The SMoE framework and attention consolidation significantly reduce computational costs while maintaining or improving accuracy.
Comprehensive Evaluation:
The method is validated on twelve cancer datasets, demonstrating its applicability and effectiveness compared to existing techniques.
Clear Problem Definition:
The paper clearly outlines the challenges in integrating histological and genomic data, providing context for the proposed solutions.
Weaknesses:

Complexity:
The framework's complexity might pose challenges for practical implementation and understanding by researchers not specialized in computational pathology.
Generalization:
While the method works well on the datasets tested, further validation on more diverse datasets could strengthen the claims of general applicability.
Comparative Analysis:
Although the paper demonstrates improved performance, more detailed comparisons with specific state-of-the-art models could provide deeper insights into the advantages of CancerMoE.
Technical Details:
Some technical aspects, such as the specific implementation of the dynamic patch selector, could be elaborated for reproducibility.

---

> ### Author Response · Authors · 2024-11-20
> **Response to Reviewer R9KF**
>
> **<Cons1> The framework's complexity.**
>
> Thanks for the remaining, we have provided the code of our CancerMoE in our supplementary and we will release our code to the public after accepted.
>
> **<Cons2> The separate neural network of DPS.**
>
> Our DPS actually does not need any separate neural network to obtain token scores.
> First, it's important to clarify that the "network" in DPS refers to the attention network from our network backbone. Specifically, we use the attention map from this network to extract the attention scores as token scores, eliminating the need for a separate neural network to assign scores to each token. The only notable aspect is that we insert a unique learnable token before each multimodal input token sequence. The sequence length of our multimodal input is over 2000, so adding one special token into the sequence only involves a slightly addition computation cost.
> Meanwhile, we have explained how to obtain the token score in Section 3.2. Thank you for pointing this out; we will emphasize this part in our revision.
>
> **<Cons3> Genomic profiles encoder and fusion method**
>
> We encode the molecular information (the genomic profiles) using "PatchEmbedding." Specifically, we treat the genomic profiles as a single vector, split them into $g$ sub-vectors, and project each sub-vector into the embedding space using a linear layer. We then concatenate the genomic profile token sequence with the image tokens to form one input sequence. This combined sequence is processed by the transformer backbone, where the self-attention modules fuse the two modalities. Thank you for pointing this out; we will include an illustration of the genomic profile encoder and the fusion details in our revision (Section 3.2).
>
> **<Cons4> Lack of multimodal baselines.**
>
> To address the reviewers' concerns, we have added a recently released multimodal baseline designed for the TCGA benchmark: CMTA (as suggested by reviewer wixR). We compare it with our method in Table 1 and in the comparison below. The results show that CMTA outperforms CancerMoE in LUSC and BRCA. However, CancerMoE still outperforms CMTA in the other 10 cancer types and maintains the best average performance overall.
>
> **<Cons5> The clarification of the BRCA performance.**
>
> First, CancerMoE achieves the highest average performance, outperforming the baselines in six types of cancer, which demonstrates its effectiveness in multimodal survival prediction. Additionally, the decline in performance for BRCA when the genomic profile modality is excluded underscores the importance of incorporating genomic profiles in CancerMoE. The performance of CancerMoE on BRCA suggests that more advanced methods are needed for this type of cancer to enhance future survival prediction.
>
> **<Cons6> Typos and Attention Map clarification.**
>
> Thanks for pointing out the typos, and we have fixed these in our revision.
> For formulate (3), we add pseudocode about ACS in our revision to improve the representation of our proposed ACS method.
> For the attention map, we show the additional visualization image in Figure 4. The visualization results show that CancerMoE can identify irregular regions and utilize this information for survival prediction.
>
> **<Cons7> Expression Improvement.**
>
> To further improve the clarity of CancerMoE, we have added pseudocode for ACS in our revision, thanks to the suggestion from reviewer wixR. We also provide the code of CancerMoE which includes the details of DPS (file: datasets\dataset_survival_new.py, line 496).

---

### Review · Reviewer_Uiia · 2024-11-05

**Summary Of Contributions:**

This paper proposes a lightweight Sparse Mixture-of-Experts model designed for efficient cancer prognosis prediction, leveraging both pathology image data and genomic data. Specifically, the authors introduce a dynamic patch selection algorithm, a sparse Mixture-of-Experts framework, and an optimized attention mechanism that consolidates and prunes redundant attention heads. These innovations reduce model memory requirements, improve training efficiency and multimodal collaboration, and enhance the model’s interpretability and overall efficiency. Experimental results demonstrate that the proposed modules effectively strengthen the model’s attention to histopathology images, facilitate information fusion between pathology images and genomic data, and achieve high cancer prognosis prediction accuracy with efficient inference.

**Audience:**

Yes

**Claims And Evidence:**

No

**Requested Changes:**

Please see the weakness.

**Strengths And Weaknesses:**

Paper Strengths
1. The SMoE model proposed by the authors can efficiently fuse pathology images and genomic data and obtains state-of-the-art performance on the cancer prognosis prediction tasks.
2. The authors propose Dynamic Patch Selection method can effectively select the critical regions in pathology images thus reducing the resource consumption.
3. The authors tailor the proposed sparse mixture expert to disentangle the model parameter space and mitigate gradient conflicts.

Paper Weaknesses
1. In the comparison experiments shown in Table 1, only two models incorporate both pathology images and genomic profiling data, while the remaining ten models rely on unimodal data. This indicates a need for additional comparative experiments with other multimodal models to provide a more comprehensive evaluation.
2. In section 4.2, the authors claim that genomic profiling may negatively affect two cancer types, LIHC and BRCA. Could you provide additional explanations or conduct further ablation studies to substantiate this claim?
3. Table 2 presents the size, inference speed and training time of the proposed CancerMoE model. However, in conjunction with Fig. 2, it appears that these metrics do not account for the resources utilized by the Dynamic Patch Selection method. Please provide detailed resource consumption data separately for the Dynamic Patch Selection method and the CancerMoE model.
4. The ablation experiment results suggest varying optimal values for the number of selected tokens, the number of Key tokens, and the ratio p across different datasets. How did the authors determine these parameter settings for the comparison experiments presented in Table 1?
5. Could the authors provide further insights into the visualization presented in Fig. 4? Specifically, an in-depth explanation of any observable patterns within the selected tokens and attention distributions would be valuable. For instance, are there recurring trends or consistencies in how tokens are selected across different data samples or categories, and how do these patterns influence the model's interpretability or performance?

---

> ### Author Response · Authors · 2024-11-20
> **Response to Reviewer Uiia**
>
> **<Cons1> More multimodal model baselines.**
>
> We have selected 12 different types of baselines for comprehensive compression. In the cancer survival prediction benchmark, both single-modal and multimodal methods keep developing, the single-modal model can also surpass multimodal baselines, as we show in Table 1, the best performance in BRCA, LUAD, and UCEC is the pathological image-only model.
> Therefore, we involve single modal baselines for the compression.
> To further address reviewer concerns, we include an additional multimodel baseline: the CMAT (ICCV'32, suggestion from reviewer wixR), and report it in our revision (Table 1).
>
> **<Cons2> Clarify: “genomic profiling may negatively affect two cancer types, LIHC and BRCA.”**
>
> This statement is based on the observation that, for these two cancer types, most genomic-based baselines perform worse than those using only pathological images. Additionally, on these two cancer types, the multimodal baselines also perform worse than most baselines which use only pathological images. Thanks for the reviewer’s reminder, in our revision, we have revised it to:
> “It indicates we need a more comprehensive fusion mechanism to effectively integrate genomic profiling with histological image in LIHC and BRCA.”
>
> **<Cons3> Resource consumption of CancerMoE.**
>
> In our paper, we report the average parameters, FLOPs, VRAM usage, and training time of the CancerMoE model. Regarding time consumption, the DPS module does not use GPU resources because it operates on the data load in parallel with model inference and training. For memory usage, the DPS uses the attention scores from the network as the token score (intrinsic in the transformer network backbone) which does not generate additional GPU memory consumption. The DPS only needs to store one FP32 number for each token (the token score, and if the token never be used, its default value is 0) in CPU memory which results in an average memory cost of $0.22$ MB per instance, which is very little (one histological image may occupy over $500$ MB).
>
> **<Cons4> Hyperparameter setting for final experiments.**
>
> In our experiments, all cancer types except for BRCA, LUSC, and SKCM use the same default hyperparameters: 3072 selected tokens, 4 key tokens, and a ratio of 0.5 for selecting unseen tokens.
>
> **<Cons5> Further insights into the visualization presented.**
>
> To address the reviewer’s concerns, we add additional visualized attention to WSIs in Figure 4 to improve the expression of our visualization. From the visualized results, we can observe that CancerMoE can identify irregular regions in WSIs which is critical for survival prediction.

---

### Author Response · Authors · 2024-11-20
**General Response**

We thank the reviewers for their time and thoughtful feedback. We're pleased that the importance of our new model for cancer survival prediction on various cancer types has been recognized. We appreciate the positive feedback on the "efficiently fuse pathology images and genomic data" (Reviewer Uiia) and the "significantly reduce computational costs while maintaining or improving accuracy" (Reviewer R9KF). Reviewers highlighted CancerMoE as a key contribution, acknowledging our model in survival prediction across the dimensions of approach, efficiency, and evaluation (Reviewer R9KF, wixR, and Uiia) and the modality gradient conflict reduction by Sparse Mixture-of-Expert (Reviewer Uiia). Lastly, we are glad our clear outlines of the challenges in integrating histological and genomic data, provide context for the proposed solutions (Reviewer Uiia).

We will now address individual questions from each reviewer below. We have also uploaded a revised version of the paper to address most of the reviewers’ concerns with the changes highlighted in red font.

---

### Decision · Action_Editor_ybGR · 2024-12-22

**Recommendation:** Reject

**Comment:**

Please see above in "Claims and Evidence".

**Audience:**

Yes.

**Claims And Evidence:**

This paper proposes a lightweight Sparse Mixture-of-Experts model designed for efficient cancer prognosis prediction, leveraging both pathology image data and genomic data. Specifically, the authors introduce a dynamic patch selection algorithm, a sparse Mixture-of-Experts framework, and an optimized attention mechanism that consolidates and prunes redundant attention heads. These techniques reduce model memory requirements, improve training efficiency and multimodal fusion.

After the rebuttal, two reviewers still have negative perspectives against the manuscript. Besides the reviewers' concerns, the AC also has the following comments regarding the paper:

- The writing of the paper needs significant improvements. For instance,
    - The authors shall delete the periods in all sub-section titles. Section 4.2 caption is inappropriate. Terms such as "powerful" and "fierce" are strange to see in a paper.
    - The using of "PatchEmbedding" in Section 3.2 isn't informative, as it's not a commonly used term.
    - The authors shall pick terms from the NLP field more.
    - The use of algorithm formats of Algorithm 1 & 2 is inappropriate as those are just some oridinary network work operations. It's very rare to see people put network operations into the algorithm format.
    - "Optimization Object." -> "Optimization Objectives".
    - There are in the paper and pleae carefully proof read the paper.

- Important details regarding the inference are missing. As mentioned in Section 3.3:
    > " Subsequently, the chosen patches are collaboratively used for the online training of our proposed CancerMoE framework"

    How the proposed design is used for inference is unclear.

- The direct comparison with sparse / dynamic ViT shall be compared (altough they are not directly designed for your task) including  Rao
et al. (2021), Kong et al. (2021). You cannot justify your design without comparing with existing and similar works.